# Evaluation of Lateral Radar Positioning for Vital Sign Monitoring: An Empirical Study

**DOI:** 10.3390/s24113548

**Published:** 2024-05-31

**Authors:** Lars Hornig, Benedek Szmola, Wiebke Pätzold, Jan Paul Vox, Karen Insa Wolf

**Affiliations:** 1Fraunhofer Institute for Digital Media Technology IDMT, Oldenburg Branch for Hearing, Speech and Audio Technology HSA, Marie-Curie-Straße 2, 26129 Oldenburg, Germany; benedek.szmola@idmt.fraunhofer.de (B.S.); wiebke.paetzold@idmt.fraunhofer.de (W.P.); jan.paul.vox@idmt.fraunhofer.de (J.P.V.); karen.insa.wolf@idmt.fraunhofer.de (K.I.W.); 2Medizinische Physik, Carl von Ossietzky University of Oldenburg, 26046 Oldenburg, Germany; 3Department of Neurology, School of Medicine and Health Science, Carl von Ossietzky University of Oldenburg, 26046 Oldenburg, Germany

**Keywords:** radar, frequency-modulated continuous wave (FMCW), sleep monitoring, vital signs, breathing rate, heart rate

## Abstract

Vital sign monitoring is dominated by precise but costly contact-based sensors. Contactless devices such as radars provide a promising alternative. In this article, the effects of lateral radar positions on breathing and heartbeat extraction are evaluated based on a sleep study. A lateral radar position is a radar placement from which multiple human body zones are mapped onto different radar range sections. These body zones can be used to extract breathing and heartbeat motions independently from one another via these different range sections. Radars were positioned above the bed as a conventional approach and on a bedside table as well as at the foot end of the bed as lateral positions. These positions were evaluated based on six nights of sleep collected from healthy volunteers with polysomnography (PSG) as a reference system. For breathing extraction, comparable results were observed for all three radar positions. For heartbeat extraction, a higher level of agreement between the radar foot end position and the PSG was found. An example of the distinction between thoracic and abdominal breathing using a lateral radar position is shown. Lateral radar positions could lead to a more detailed analysis of movements along the body, with the potential for diagnostic applications.

## 1. Introduction

Contact-based measurements are the standard method for vital sign detection both at home and in medical facilities. A contact sensor offers advantages in precision and validity based on its firm placement at a known body position. These advantages come with the drawbacks of disinfecting, applying, and adjusting the sensor as well as the constant restraint on the human body due to the sensor. Contactless sensors provide a promising alternative, avoiding these drawbacks of contact solutions while providing comparable quality [1,2]. There is ongoing research in this field to develop various contactless methods for detecting vital signs, including the use of audio, thermal imaging, or depth sensors in different measurement setups [1].

This publication focuses on the use of a frequency-modulated continuous wave (FMCW) radar for the derivation of breathing and heart rate as important vital parameters during sleep. Continuous observation of these two vital signs and the identification of sleep disturbances are relevant use cases for health monitoring. From a technical point of view, sleep monitoring is an advantageous context for contactless sensors, as large body motions are rare during sleep, and cannot mask the breathing and heartbeat pulse motion. Yet, even for sleep monitoring, there are prominent challenges in the radar vital sign detection field, such as masking of the heartbeat pulse by breathing motion [3,4]. Current algorithmic advances in this field could be supported by the new measurement approach proposed in this publication. In this approach, contactless radar-based vital sign measurements could become more viable and provide a simpler alternative to the current contact-based solutions in everyday use.

### 1.1. Radar Technology for Extraction of Vital Signs

FMCW radars modulate their transmitted frequency over time to differentiate objects in their Field of View (FoV) based on their distance to the radar. A common frequency modulation is a linear frequency change, which results in a signal called a “chirp”. Based on the frequency difference $f_{d}$ between the received and emitted waves, the speed of light *c*, and the rate *S* of frequency increase over time, the distance *d* to a static reflecting object can be derived as
(1)$$d=\frac{f_{d}c}{2S}.$$

Algorithms such as the Fourier transform can be used to derive the frequency and phase information of the chirp signal. Each frequency bin of the frequency spectrum corresponds to a segment of space in the form of a spherical shell in the radar’s FoV, also called a range bin (Figure 1). Based on the information of one receiving and one transmitting antenna, the objects in one such shell are superimposed. The depth $d_{shell}$ of such a shell is defined as
(2)$$d_{shell}=\frac{c}{2B},$$
where *B* is the frequency bandwidth of the swept frequency for a chirp. A possible bandwidth for commercially available radars is 3 GHz, leading to a shell depth $d_{shell}$ of about 5 cm.

**Figure 1 sensors-24-03548-f001:**
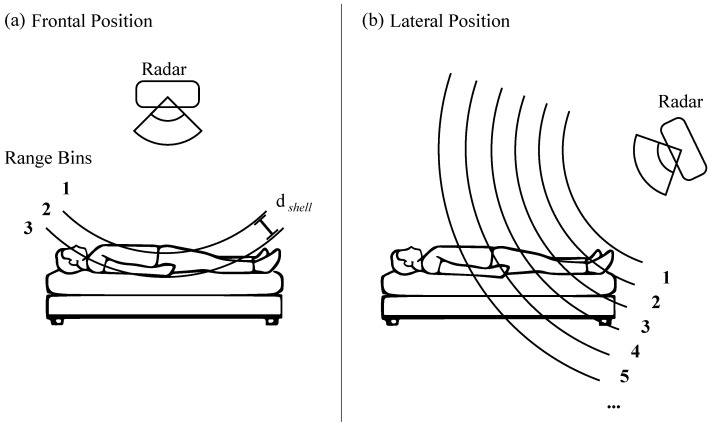
Two different radar positions for vital sign extraction in a sleep context. The radar with a top-down view in (**a**) depicts the entire body in a small number of range bins, whereas the radar with a lateral-view in (**b**) maps multiple human body zones onto many more range bins.

In addition to this rough distance information, the phase information in each range bin can be evaluated to obtain a much finer resolution of the distance. By analysing the phase of the reflected wave, even small motions, such as the torso motion during breathing and the heartbeat pulse, can be detected. The phase φ can be expressed as [5]
(3)$$\varphi =\frac{4\pi f_{min}R}{c},$$
where $f_{min}$ is the minimal frequency of the FMCW radar with an increasing frequency, *R* is the displacement, and *c* is the speed of light. Paterniani and colleagues provided a more detailed explanation of the details of the FMCW signal and its processing in [6].

For a person in front of a radar, inhalation causes the torso to inflate and move towards the device. The electromagnetic wave is reflected earlier in its period compared to the neutral state of the torso. This shift in phase can be detected by the radar and used to derive the characteristic body motions associated with vital signs. The heartbeat can be detected in a similar manner. The heart contractions vibrate the chest and send a pulse along the body’s blood vessels, making them expand. This vibration and expansion can in turn be measured by the radar. In humans, the heartbeat pulse motion has a much lower amplitude compared to the breathing motion. The displacement of the whole breathing motion is in the range of about 1 mm to 5 mm across the torso [7], while the skin displacement in the area above the heart during the heartbeat pulse has been reported to range from 0.287 mm to 0.568 mm [8].

The phase-based range accuracy in one range bin of the FMCW radar is less than 1 mm, placing it in the same motion amplitude range as the breathing motion and heartbeat pulse [9,10]. Bhutani et al. provided a detailed investigation of the range accuracy of a 60 GHz FMCW radar similar to the one used in this study in [11].

### 1.2. Related Work

The masking of the small heartbeat pulse motion by the large breathing motion is a critical problem for radar-based vital sign detection. The smallest displacement caused by breathing is about 1 mm. This is almost twice as large as the largest displacement caused by a heartbeat of about 0.57 mm reported above. In the time domain, the large breathing motion can hide the heartbeat pulse motion. Similarly, in the frequency domain, large breathing motions with a frequency of up to 0.4 Hz [12,13] lead to harmonics that reach into the heartbeat frequency range of 0.83 Hz to 3.66 Hz [14,15]. This spectral overlap is especially prevalent in sleep, as resting heart rates in this state are in the lower part of the relevant frequency range. The non-stationary characteristic of breathing and heartbeat frequencies result in the fact that longer observation windows do not necessarily solve this signal separation problem, as the overlap in heartbeat pulse and breathing motion is an inverse problem. Current approaches to solve this issue use either complex calculations or advanced hardware setups. Oshim et al. used Gabor filters of different spatial wavelengths together with machine learning to magnify the small heartbeat pulse motions [3]. Lin et al. used a specifically tailored radar with two frequency bands, ensemble empirical mode decomposition, and principal component analysis (PCA) to extract the heartbeat signal from the respiratory interference [4]. Hasan et al. presented a comparison of eight different algorithms for detection of vital signs with radar [16].

A smaller number of publications have used multiple range bins which span the human body, and have compared the phase information of multiple receiving and transmitting antennas in order to calculate the incidence angle of the reflected electromagnetic waves. Upadhyay et al. used this range and angle information to find the position of the most prominent heartbeat pulse along a sitting human [17]. Li and colleagues tried a similar approach to find vital signs along the lying body of a human [18].

All of these algorithms for vital sign extraction have led to good results for breathing and promising results for heartbeat. Nonetheless, they are either computationally expensive algorithms or require high-performance hardware. Both aspects might present disadvantages in the development of a medical product. In such a setting, cost-efficient hardware and simple explainable algorithms are beneficial. Additionally, a complex algorithm can break in complex ways if unforeseen circumstances are encountered. Large body motions from limbs or pathological vital signs could represent such circumstances. To achieve more robust vital sign extraction, an improvement of the initial measurement situation could be a promising step.

### 1.3. Proposed Approach for Radar Positioning

We propose the intentional use of a lateral radar position for separating breathing motion and heartbeat pulse. A lateral radar position is a radar placement from which multiple human body zones are mapped onto different radar range bins (see Figure 1b). Following this, these body zones can be observed independently over time via these range bins. Body zones exhibit human vital signs in different ways. The breathing motion is visible at the torso, but is absent at the legs, while the legs still exhibit a heartbeat pulse. This inherent separation of vital signs traces along the body is present in the range bins of a lateral radar position as well. Therefore, the lateral positioning of the radar offers a different starting point for the separation of the two signal components. No inverse problem needs to be solved to extract the heartbeat motion.

The scientific novelty of this publication lies in its examination of how the lateral position of an FMCW radar combined with physiological assumptions can inherently contribute to the solution of the present problem of vital signs being masked. All radar vital sign measurements that place the radar in the standard way in front or behind the torso lead to large body zones being present in a small number of range bins. This in turn leads to an overlap between the breathing motion and heartbeat pulse within these range bins, as well as to fewer range bins that can be investigated for details of vital sign motions (see Figure 1a). These measurements cannot use the effect described above. Only a small number of publications have investigated the selection of multiple range bins or a lateral radar position for vital sign separation (two examples [17,18]), and these did not intentionally use the different physiologically-based characteristics of the body movements in the different body zones. To the best of the authors’ knowledge, no publication has yet explored this. Merely using a lateral radar position without any further assumption about the body neither exploits the advantages of a lateral radar position in the vital sign measurement context nor considers the disadvantages of such positioning.

Advantages of the approach using a lateral radar position combined with physiological assumptions around body motions, from here on called “informed lateral radar position”, can be illustrated by the radar position at the bed foot, as seen in Figure 1b. Here, the human body is mapped onto the range bins, starting from the feet close to the radar and ending at the head further away from the radar. The heartbeat pulse should be easier to detect, as it is already inherently separate from the breathing motion during the measurement. An approach considering human physiology would advise against searching for the best heartbeat pulse only in range bins close to the breathing motion due to the possibility of overlapping motions. In the same measurement setup, the spatial separation of the human body along the range bins could be used to extract additional information, such as separating the thoracic and abdominal breathing. Comparative analysis of these two regions’ breathing signals could prove useful in detecting paradoxical breathing patterns, which are indicators of both obstructive sleep apnea (OSA) [19,20] and chronic obstructive pulmonary disease (COPD) [21]. If this were shown to be true, the detection of other small motions along the body, such as muscle tremors [22] or the heartbeat pulse wave along the range bins, could also be possible. These are only a few examples illustrating how an informed lateral radar position could possibly help to extract more medically important signals.

To explore these hypothesized advantages, this publication focuses on six nights of sleep measurements using the standard radar position and a lateral radar position similar to the one shown in Figure 1a,b. The differences between radar positions are quantified against contact-based reference sensors using the vital sign detection rate, reliability over time, and accuracy of the calculated vital signs. A better match between the detected radar vital signs and reference sensors is considered an improvement in this context. Following the described effects, the informed lateral radar position should not suffer from masking of the heartbeat pulse by the breathing motion. This should lead to a better heartbeat pulse detection for the lateral radar position compared to the standard radar position. The breathing detection should be comparable for the informed lateral radar position compared to the standard radar position, as breathing detection is not a problem in the current radar vital sign detection literature. Thus, the prominent breathing motion should remain extractable even if the breathing movement is reduced in the lateral view compared to the top-down view.

## 2. Materials and Methods

### 2.1. Experimental Setup

The proposed approach was tested by performing a sleep study. Radar devices were placed at three different positions around the bed, as shown in Figure 2. Radar 1 was positioned at the foot end of the bed, approximately 1 m above the surface of the mattress, radar 2 was positioned on the bedside table at approximately the height of the mattress, and radar 3 was positioned vertically 2 m above the bed.

In parallel, the subject was monitored with the contact-based sensors of a polysomnography system (PSG) as reference; cf. Section 2.4. PSG is the standard diagnostic tool in sleep medicine for assessing human sleep [23].

Position 3 represents a commonly used radar position in literature, as described in Section 1.2. From this radar position, the human body zones should be summarized into a small number of range bins, and the breathing motion and heartbeat pulse should overlap in all of them. Masking of the heartbeat pulse by the breathing motion is present with this positioning. Starting from the foot end, Position 1 distributes the different human body zones over many range bins, which fulfills the separation of the breathing motion and heartbeat pulse described in Section 1.3. Additionally, the human body zones can be assigned to the range bins more easily, as this foot end radar position has the feet closer to it by definition. This simplifies the examination of individual body zones as signal sources during informed usage of the lateral radar position. Position 2, on the nightstand, should also have the different human body zones mapped onto multiple range bins, enabling the described measurement-based separation of breathing motion and heartbeat pulse. This position was added as an alternative lateral position to Position 1.

### 2.2. Subjects

Eleven adults (four females, seven males, mean age = 32.7 years, range = 19–45 years) participated in the study. To be a part of the study, the subjects had to be healthy adults capable of providing informed consent. All eleven participants were self-reported to be healthy and were not diagnosed with sleeping disorders. However, some time after the completion of data collection, one participant informed us about self-reported sleep apnea symptoms. Written and informed consent was provided by all participants. The study was reviewed and approved by the Kommission für Forschungsfolgenabschätzung und Ethik of the Carl von Ossietzky University of Oldenburg (protocol number: Drs.EK/2021/079-01, date of approval: 7 September 2022).

Two datasets were excluded from the analysis because one radar sensor failed to measure during the night, and three additional datasets were excluded because the PSG and the radar systems could not be synchronized afterwards. Datasets from six participants (subjects 06–11) were ultimately used in the analysis.

### 2.3. Radar Configuration

The radar used in the setup was a 60 GHz FMCW radar IWR6843ISK-ODS from Texas Instruments (Dallas, TX, USA). Table 1 shows more information on the radar and the configuration used in this study. This radar was chosen because of its large field of view in azimuth and elevation to cover the whole body of a person from a lateral view. The radar measurement boxes highlighted in Figure 2 were custom-made and included an IWR6843ISK-ODS radar board, real-time DCA1000EVM measurement board from Texas Instruments, and Raspberry Pi 4B that transmitted the radar data to a laptop via WiFi. On the laptop, the Lab Streaming Layer (LSL) protocol was used to save and synchronize the data of all three radars to one timeline [24].

The sampling frequency of the radar was set to 20 Hz, which leads to a Nyquist limit of 10 Hz. The maximum heart rate is the highest expected frequency of the goal vital signs, at 3.66 Hz [14,15]. This 3.66 Hz heart rate is well below the Nyquist limit of 10 Hz.

### 2.4. Reference Sensor System

As ground truth for the radar vital signs, the “SOMNOscreen plus” (SOMNOmedics, Randersacker, Bavaria, Germany) PSG system was used. It comprises many different contact-based sensors, all linked to a recording box worn by the subject on a belt around the upper body. The sensors used for data recording are listed in Table 2. In this study, only a select few of these sensors were evaluated. For breathing, the thoracic and abdominal motion was measured by piezoresistive and respiratory inductive plethysmography (RIP) belts. For the heartbeat, the electrical voltage was measured by electrocardiography (ECG) electrodes.

### 2.5. Experimental Protocol

The overnight measurements were performed in an office building room. The bed had a size of 1.4 m by 2 m (Figure 2). The radars were placed around the bed as described in Section 2.1 and were aligned towards the center of the bed.

At the start of the measurement procedure, the participants were asked to put on their sleepwear. Afterwards, the PSG sensor system was attached and recording by the PSG system and radar sensor system were started. The participants could then lie down in bed. During the night, the participants could move freely in the bed and temporarily leave the room. The recording was not stopped during these events. After waking up, the participants could detach all sensors and leave.

### 2.6. Measurement Principles of the Different Sensors

To construct algorithms for deriving vital signs from the sensors and to compare their results, the differences between the sensors themselves need to be considered. For radar vital sign detection, the phase information of the electromagnetic waves reflected from the skin is used for this. These waves penetrate regular bed sheets and clothing, then are mostly reflected by the water contained in human skin. During breathing, the distance between the radar and the human torso changes cyclically as the torso inflates and deflates. The heartbeat is detected by the radar via the pulsation of the surface vessels as blood is pushed through them by the contraction of the myocardium. When interpreting the radar phase signals, a rising value depicts the object moving further away from the radar; conversely, a decreasing phase value corresponds to the object moving closer to the radar.

From the PSG sensors (SOMNOmedics, Randersacker, Bavaria, Germany), we took the piezoresistive belt wrapped around the thorax of the participant as the breathing reference. The sensor signal is directly proportional to the breathing motion; a rise of the sensor value corresponds to an increase in the torso volume, whereas a decrease corresponds to a reduction of the volume. Therefore, the PSG and the radar breathing signals essentially represent the same phenomenon, that is, the cyclical inflation and deflation of the torso. While the piezoresistive belt is directly connected to the torso as its signal source, in the radar algorithm a human body needs to first be identified in the radar’s FoV. Additionally, it must be considered that the amplitude of the breathing motion in a range bin depends on the perspective of the radar relative to the torso. Thus, for radar vital sign detection, an additional step is required to first identify the best prominent breathing signal in the radar data.

The reference heart rate value was obtained from the 2-lead electrocardiogram applied to the participants thorax. The electrical potentials measured by the ECG drive the heart’s pumping activity, which in turn causes blood to flow through vessels throughout the body. The resulting change in diameter of these vessels is what can be seen in the radar signal. This relationship introduces a delay between the prominent ECG cycle and the radar heartbeat pulse. This delay also changes based on which body zone is observed by the radar. As mentioned above, as an additional step it is also necessary to find the human heartbeat pulse in the field of view of the radar.

### 2.7. Vital Rate Extraction Algorithms

Vital rates were derived using a sliding analysis window with a window length of 60 s, with an overlap of 40 s between consecutive windows to achieve adequate frequency and time resolution. For each window, the algorithms described below return one average breathing and heartbeat frequency, expressed in breaths or beats per minute (BPM), along with the range bin in which the breathing and heartbeat signals were identified. Based on the frequency resolution resulting from the chosen window length, differences in vital rates between the sensors can be distinguished down to 1 BPM. When designing the algorithms, in addition to accuracy, computational simplicity was an important aspect. As described in Section 1.3, part of our hypothesis was that complex algorithms for the separation of breathing and heartbeat signals would not be necessary when using the novel radar positioning. All of the algorithms were run separately on each included subject’s data. The resulting radar and reference vital rates were then combined to compute the results.

#### 2.7.1. Range Bin Selection

As mentioned when describing the sensor principles in Section 2.6, before computing vital rates from radar signals, a fitting range bin has to be identified. We developed an algorithm that is applicable for both breathing and heartbeat analysis with only a few adjustments. The steps of the algorithm and their respective considerations are as follows:
(a)The spatial range where potentially informative signals can be found is known, being between 0.45 m and 3 m. The lower bound is set because of the radar antenna cross-talk effect between sending and receiving antennas, which becomes prominent below this range. The upper bound is set by the bed length. Therefore as a first step, data from range bins outside of the 0.45–3 m range are discarded.(b)Range bins exhibiting excessively large sample-to-sample phase difference values are discarded. Such phenomena can be generated by either artifacts or large body movements, both of which hinder robust and accurate vital rate extraction. We selected a threshold value of 1 mm sample-to-sample difference. This selection was guided by the exploration of the recorded dataset and the knowledge that the average breathing displacement range is between 1 and 5 mm (cf. Section 1.1 and [8]). The change in displacement in the 50 ms between two radar samples should not be as large as a whole breathing motion. By evaluating this threshold test separately for each range bin, we take into account possible situations where only one part of the body is moving while the others are relatively calm, allowing vital sign data to be extracted in these cases. If all range bins fail this test, no vital signs are extracted for the given time window, and it is marked as having no radar vital signs.(c)It has been shown that the temporal variation of the reflected wave’s magnitude is a good measure for distinguishing human subjects from their stationary surroundings [25]. Stationary objects should reflect the same amount of energy throughout the whole measurement, whereas humans constantly exhibit some degree of movement, meaning that their reflected energy varies. The standard deviation is used to quantify this variation for each range bin for the analyzed time window. The bins with the highest standard deviation are selected together with their surroundings. This step is only applied when looking for breathing signals, as range bins showing the heartbeat pulse often do not stand out by their standard deviation because of their lower amplitude (cf. Section 1.1 and [8]).(d)The remaining range bins are then analyzed for the presence of regular periodic activity using a measure called the temporal phase coherency (TPC) [26]. To compute the TPC, the phase signal is first subjected to a bandpass filter with a second-order Butterworth filter. The filter ranges are 0.1–0.4 Hz for breathing and 0.8–1.7 Hz for heartbeat [26]. The filtered signal is then normalized by its envelope, resulting in a signal bounded in (−1,1), and the envelope is computed using the Hilbert transform. Using the normalized signal, the average period length is computed based on the number of zero crossings. The coherency value is then calculated between the normalized signal and a version of itself that is shifted by the average period length. The formulaic expression of the TPC for a given time *t* and range bin *r* is as follows:
(4)$$TPC\left(t,r\right)=\frac{\sum_{s=t-t_{0}+T}^{t}\;\hat{P}\left(s,r\right)\cdot \hat{P}\left(s-T,r\right)}{\sigma_{\hat{P}}\left(t,r\right)\cdot \sigma_{\hat{P}}\left(t-T,r\right)}$$
where $\hat{P}$ is the normalized phase signal, *T* is its average period length, and σ is its standard deviation [26]. The five range bins with the highest TPC values then proceed to the vital rate computation algorithms.

#### 2.7.2. Breathing Rate

Considering the consensus between the representation of breathing in the reference sensor signal and radar sensor signal, the breathing rate computation was performed using an identical algorithm for both sensors.

Autocorrelation was chosen because of its sensitivity for periodically changing signals. The autocorrelation function is computed by shifting the signal along itself and computing the correlation value ρ for each shift *k*, also called the lag:
(5)$$\rho \left(k\right)=\frac{Cov(X_{t},X_{t-k})}{\sigma \left(X_{t}\right)\cdot \sigma \left(X_{t-k}\right)}$$
where Cov is the covariance, $X_{t}$ is the variable at time *t*, and σ is the standard deviation. A perfectly noiseless periodic signal produces an autocorrelogram with peaks only at those shifts corresponding to multiples of the signal’s periodic length. To make the algorithm more robust against noise and artifacts, an additional step was included. It has been reported that random body motions can be detected based on the width of the autocorrelogram’s middle peak [27]. Thus, we defined the width as the lag difference between the wave troughs surrounding the middle peak at lag 0. If this width is outside of the normal breathing cycle length, then the rate computation is stopped. If a range bin passes the autocorrelogram peak width test, then the rate computation resumes. We first detected the peaks of the autocorrelogram in the lag range corresponding to the physiological breathing rate range, then took the inverse of the first peak’s lag value to compute the breathing rate. In order to decide between the five range bins provided by the TPC step, the correlation value of the autocorrelogram’s first peak and the standard deviation of the inter-peak distances were used. A smaller standard deviation and higher correlation value is evidence of a more consistent and clearer signal.

#### 2.7.3. Heart Rate

As the reference sensor signal and radar signal represent different phenomena (cf. Section 2.6), their resulting heartbeat signals are also quite different (see Figure 3). Thus, it is necessary to use separate algorithms to compute the heart rates from the PSG and the radars.

For the PSG ECG, a method based on the wavelet transform (WT) with the Symlet 4 wavelet was used. This wavelet has a similar structure to the characteristic ECG QRS complex, making it an appropriate choice for its extraction [28]. The ECG signal was decomposed using the stationary WT algorithm, and the component corresponding to the typical pulse frequency was used for the next steps. Afterwards, an envelope of this component was computed and its peaks were detected. The heart rate was derived by taking the inverse of the average peak-to-peak interval.

The radar heart rate computation starts with the normalized signal created in the TPC computation step (cf. Section 2.7.1). The peaks of this signal were detected and the peak-to-peak intervals were calculated. A median filter was applied to these intervals to lower the effect of missed heartbeats. If the average of the intervals was outside the physiological range, the bin was discarded. The best bin from the five selected via TPC was the one with the lowest standard deviation of the peak-to-peak intervals. Finally, the heart rate was computed by taking the inverse of the average median filtered inter-peak interval.

## 3. Results

We analyzed 33.05 h of radar and PSG data. This represents six nights of sleep, with one night for each of the six included subjects. The presented results originate from the combined analysis of all datasets, except in cases where we have stated that certain subjects were excluded.

### 3.1. Breathing Rate

Using the described algorithm for the reference PSG thorax belt (cf. Section 2.7.2), the overall ratio of time windows with detected breathing activity was 93.9%. The results of the radar-based breathing rate are listed in Table 3. For the radars, the values were as follows: foot end, 98.5%; nightstand, 97.5%; ceiling, 81.4%. Subject 11 had no time windows of detected breathing with the ceiling radar. Without subject 11, the ceiling radar’s detection rate was 97.8%. To quantify the reliability of breathing signals for the different radar positions, we computed the proportion of time windows for which the difference compared to the PSG was lower than ±3 and ±1 breath(s) per minute. We only counted windows where both the radar and PSG detected breathing. Each radar had a proportion between 98% and 99% for ±3 breaths per minute and between about 93% and 94% for ±1 breaths per minute. To investigate the accuracy of the radar-based breathing rate in comparison to the reference, we computed the Mean Absolute Percent Error (MAPE), with the results shown in Table 3. The MAPE values were all below 2.5% and within 0.4% of each other.

Considering that multiple body zones are mapped to different range bins with the lateral radar position, we investigated whether it is possible to distinguish between thoracic and abdominal breathing based on data of the radar at the foot end of the bed. Separate recordings were performed with a test subject simulating thoracic and abdominal breathing. The resulting breathing range profiles are shown in Figure 4. While visual inspection revealed no noticeable difference in waveform, a different spatial distribution of the breathing signal across the range bins was apparent. The algorithmically chosen range bins, marked in red, correspond to the type of breathing, with the chosen bin for thoracic breathing being further away than for abdominal breathing (1.65 m and 1.5 m, respectively).

### 3.2. Heart Rate

Based on the ECG detection algorithm (cf. Section 2.7.3), a heart rate was derived in 92.6% of the time windows. For subject 11, only 62.3% of the ECG time windows resulted in a detected heart rate. By excluding this subject, the overall reference sensor detection rate changed to 98.7%. The results of the radar based heart rate are listed in Table 4. For the radars, the values were as follows: foot end, 70.6%; nightstand, 59.6%; ceiling, 60.0%. As the results of subject 09 for all radars and subject 11 for the ceiling radar showed substantial divergence from the others, we recomputed the ratios based on a subset in which the data of subject 09 were removed for all radars and the data from subject 11 were removed for the ceiling radar, resulting in 81.4%, 66.1%, and 82.3% success rates for heartbeat extraction. We performed analyses analogous to those used for breathing to quantify the reliability of the different radar positions in detecting the heartbeat pulse (Table 4). To account for the physiologically wider range of heart rate values compared to breathing rate values, the upper tolerance value for differences in this case was ±5 and ±3 heartbeats per minute. Based on the complete dataset including all subjects, the ratios of the extracted heart rates were within the ±5 criterion: foot end, 92.3%; nightstand, 87.7%; ceiling, 76.1%. For the ±1 criterion, the following results were reached: foot end, 58.2%; nightstand, 37.2%; ceiling, 32.2%. We recomputed the results while excluding the above-mentioned data from subjects 09 and 11. The values for the ±1 criterion improved, at 60.4%, 37.6% and 35.2%, respectively, but showed the same trend. In terms of accuracy, the MAPE values showing the match with the reference heart rates were 3.25%, 4.09%, and 6.09%, respectively. After excluding subjects 09 and 11 as described above, the results were 2.04%, 3.53%, and 4.18%, respectively.

## 4. Discussion

In this work, we propose the intentional use of lateral radar positions combined with physiological assumptions regarding body motions in order to map multiple body zones to different range bins. We compare this approach to the more commonly used radar positions directly above or below the bed. We hypothesized that a lateral position should improve the radar’s vital sign detection rate, reliability, and accuracy for heartbeat pulse extraction. The quality of the breathing detection should be similar between lateral and standard radar positioning. Additionally, more information, such as thoracic and abdominal breathing, could become accessible with a lateral radar position.

To provide a first view of the effects of a lateral radar position, data from an empirical study based on three different radar positions were analysed, along with a synchronized PSG used as a reference. The total duration of the measurements added up to 33.05 h of data. One radar was fixed to the ceiling, representing the conventional position, while two radars were placed laterally, according to our hypothesis, one at the foot end of the bed and another on a nightstand. An algorithm for the selection of the range bin with the clearest vital sign was designed. In the design process, we combined and adapted different state-of-the-art algorithms which have previously been shown to provide good results in radar-based vital sign monitoring (cf. Section 2.7.1). The detection rate of vital signs was calculated for all three radars, the PSG ECG, and the PSG chest belt. The reliability measure was the difference between the radar and the PSG sensor in the percentage of the detected vital sign rate below a defined threshold. The accuracy measure was the Mean Absolute Percent Error (MAPE).

Looking at the results of the breathing analysis, there are no large differences between the three sensor positions. Initially, the lateral positions seem to perform better (cf. Table 3); however, this difference is caused by the data from one particular subject. When the ceiling radar data for this subject are excluded, the detection rate of the ceiling radar is within 1% of the two lateral radars. We investigated what could have caused this difference for this single subject, but found no valid explanations. In terms of both reliability (time within the BPM threshold relative to PSG sensor) and accuracy (MAPE), the radars performed equally well, with each within a 2.5% MAPE error of the reference. Therefore, based on the analyzed data, the use of the proposed radar positions is valid for monitoring breathing. The results are in line with our hypothesis about radar positioning, as breathing is the dominant body motion of a sleeping human and as such should be detectable even without the spatial separation of body zones. Measuring more subjects in the future will help to determine the reasons for outlier effects in individual subjects.

Concerning breathing, further investigations were carried out to demonstrate the potential for distinguishing specific patterns of thoracic and abdominal breathing based on the proposed lateral radar positions. Using the foot end radar, a test subject deliberately performed abdominal and thoracic breathing. These two breathing modes produced a different distribution of movements along the range bins spanning the body (Figure 4). The automatically identified range bin can be allocated to the thorax and abdomen. The area spanned by thoracic breathing partially also overlapped with the area of abdominal breathing, and reached further away from the radar towards the subject’s head. The physiological explanation for this overlapping of areas is the connection between the thorax and abdomen. Even during thoracic breathing, the diaphragm remains involved, leading to a general movement along the whole torso. Further investigations should be carried out on this topic. Other pathological breathing patterns could be investigated as well.

Heartbeat pulse detection should benefit from a lateral radar position, as the relatively low amplitude signal can be spatially isolated from the large breathing motion. The proportion of time windows with detected heart rates hints at heartbeat detection being more difficult compared to breathing detection. When computing the results based on the whole dataset including all subjects, the best result was 70.6% with the foot end radar. The other two radars were close to 60%. These detection values are more than 15 percentage points lower than the breathing detection rates. We investigated the results further by excluding two of the six subjects, as their data showed clear divergence from the others. This improved the ceiling radar by around 20 percentage points to 82.3%, whereas the lateral positions, especially the nightstand radar, showed less of an improvement. More data should be collected in order to determine whether this change reflects the effect of radar position or inter-subject variability. Without subject exclusions, the best MAPE was 3.25% for the foot end radar, although the ceiling position, with the worst performance, still reached an acceptable 6.09%. The order of these measurements did not change after excluding the above-mentioned subjects; the foot end position was the best at 2.04%, followed by the nightstand 3.53%, then the ceiling 4.18%. Taken together, the heartbeat detection results after subject exclusion show an interesting phenomenon. While the detection rates of the ceiling and foot end radars are similar, within 5%, the foot end radar has a considerable advantage in the proportion of windows within ±5 BPM of the reference (roughly +13%) and especially in the proportion of windows within ±1 BPM (roughly +25%). In addition, the MAPE of the foot end radar is around 2% lower compared to the ceiling radar. This can be interpreted as the foot end radar capturing clearer heartbeat signals, as a higher proportion of the signals can be used for computing accurate heart rate values. The ceiling radar, on the other hand, captures signals which are more likely to be too unclear to derive an accurate heart rate, at least with the straightforward analytical approach presented in Section 2.7.3. In keeping with the aim of simple analysis algorithms, the proposed approach involving supporting signal separation through spatial separation based on a lateral radar position seems advantageous.

The nightstand radar performs worse than the foot end radar, especially in terms of the detection rate metric. A possible reason for this effect is that the nightstand radar may have a worse view of the legs. The legs could be important, as they are the most distant body zone from the breathing motion of the torso in which the heartbeat pulse can still be detected. As seen in Figure 2, the nightstand radar was placed close to the upper edge of the bed, approximately at the height of the human torso. From this position, the legs make up only a small part of the nightstand radar’s field of view. In contrast, the radar at the foot end was placed higher up and closer to the legs, giving it a clearer view, which could explain its better performance. This should be investigated in the future using data from more subjects.

In general, objects that are further away from the radar are captured in a smaller area of the radar FoV, and the signal attenuation of the electromagnetic wave is larger. Based on this, the SNR for vital signs of body zones further away from the radar decreases. This in turn influences the accuracy of the phase estimation [9,10]. This could have influenced the results of the ceiling radar, as it was mounted 1 m further away from the body than the radar at the foot end. Viewed the other way around, the position of the foot end radar may have benefited from a lack of breathing motion for the heartbeat extraction at the feet as well as from a magnifying effect for the small heartbeat motion from body zones closer to the radar. This effect should be quantified in the future in order to differentiate the influences of the distance and radar angles with respect to the body.

A limiting factor in the interpretation of our results is that the subjects’ presence in the bed was not controlled. According to the experimental protocol, the subjects were allowed to leave the bed during the night to go to the bathroom, in which case the radars were not able to measure vital signs while the PSG could continue to measure them. This introduces a hurdle in evaluating how often the radars would be able to detect vital signs. We took the compromise of only looking at those time windows where the algorithms found a vital sign in both the PSG and the radar signal. This was done separately for breathing and heartbeat pulse. A further limitation is that we used self-developed algorithms when computing the reference vital rates, which have not yet been independently validated; however, we did carry out extensive manual inspection of the vital rates computed from the PSG, finding that they were in good agreement with the raw data.

## 5. Conclusions

In the current analysis, we have focused on showing the potential of a novel approach for intentionally positioning a radar laterally, with the human physiology in mind, in order to map the body to different range bins and achieve spatial separation of breathing and heartbeat motions.

In summary, the lateral radar positioning is proven to be equivalent to the commonly used vertical position above the bed in the case of breathing rate extraction and better in the case of heart rate extraction when compared to reference sensors. The foot end radar position leads to an improvement in both the heartbeat detection rate and its accuracy. The lateral nightstand position should be investigated further as well, as it is possible that its worse performance was due to a misaligned perspective with regard to the body. Additional aspects, such as thoracic versus abdominal breathing and the distance from the radar to specific body zones, should be investigated further as well. The representation of thoracic versus abdominal breathing was shown to be exemplary, suggesting that this approach has additional potential for identifying specific breathing patterns. Furthermore, the simultaneous use of multiple range bins should be considered as a way of making vital sign extraction more robust against artifacts.

In conclusion, radar positioning could be a simple yet powerful tool in the toolbox for radar-based vital sign detection. An intentionally chosen position could simplify data evaluation without adding cost, while at the same time revealing more information through localization on the body or spatial separation of different signal sources. These advantages could increase the use of noncontact vital sign monitoring, benefiting both patients and caregivers alike.

## Figures and Tables

**Figure 2 sensors-24-03548-f002:**
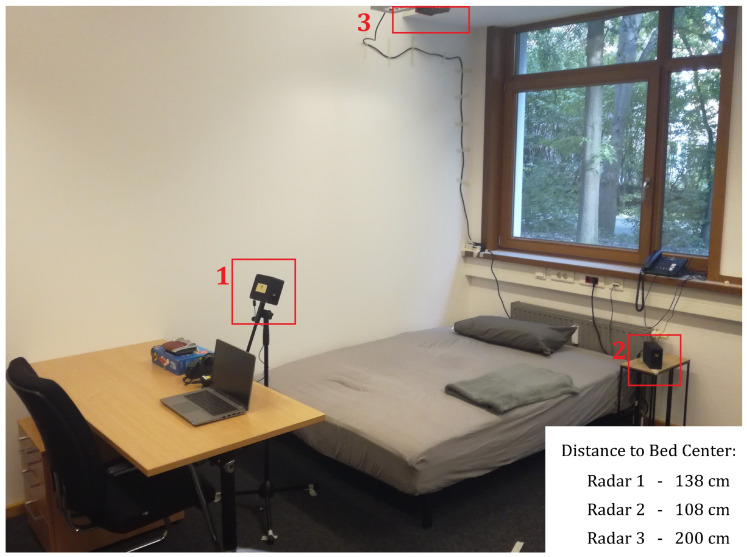
Three radars were positioned around the measurement bed and directed towards the center of the bed. Radar 1 was mounted on a stand at the foot end of the bed, Radar 2 was positioned on a nightstand at the side of the bed, and Radar 3 was fixed to the ceiling directly above the center of the bed.

**Figure 3 sensors-24-03548-f003:**
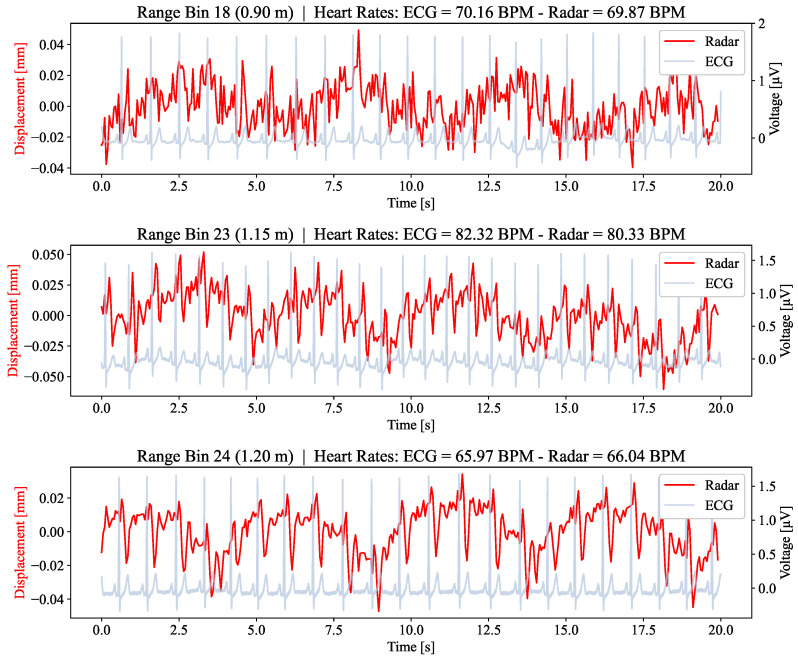
Examples of the raw phase signal in time windows with automatically detected heartbeat for low, medium, and high signal-to-noise ratios. All examples were chosen from the same participant and measured by the bed foot radar at different times and in different range bins. The heartbeat frequency is listed as beats per minute (BPM).

**Figure 4 sensors-24-03548-f004:**
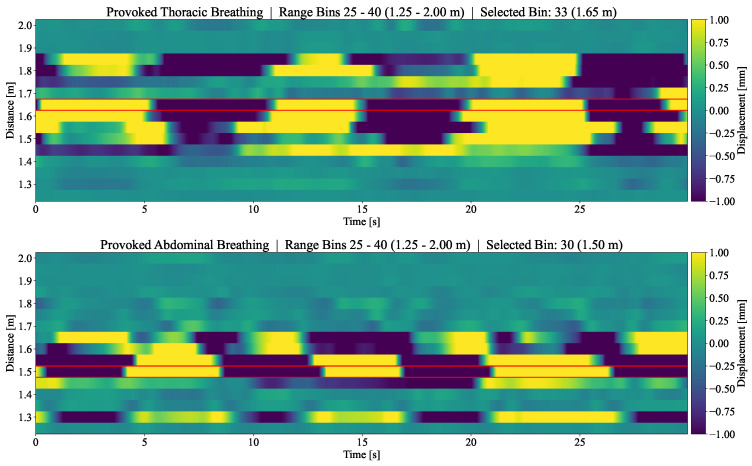
Distribution of breathing motions across multiple range bins in two test measurements with provoked abdominal and thoracic breathing. Radar 1, positioned at the foot end, was used for this experiment. The red box in each figure indicates the range bin chosen by the automatic breathing detection algorithm.

**Table 1 sensors-24-03548-t001:** Radar configuration.

Frequency band	60–63 GHz
FMCW Range resolution	5 cm
Azimuth and elevation FoV	120°
Receiving antennas	4
Transmitting antennas	1
Transmitted power in boresight	19 dBm
Time between two frames	50 ms
Number of chirps per frame	12
Idle time between two chirps	368 µs
Duration of chirps	33 µs
Samples in one chirp	128
Type of chirp	monotonically increasing

**Table 2 sensors-24-03548-t002:** PSG Configuration, SOMNOscreen Plus (SOMNOmedics, Randersacker, Bavaria, Germany).

Measured Quantity	Sensor (Sampling Frequency)
Breathing motion	Abdomen and thorax RIP belt (32 Hz)
Breathing motion	Piezoresistive thorax belt (32 Hz)
Airflow through nose	Nasal cannula (256 Hz)
Electrical heartbeat trace	ECG (2 electrodes, 256 Hz)
Oxygen saturation (S_P_O_2_)	Finger clip (4 Hz)
Leg muscle activity	EMG on lower legs (2 electrode, 128 Hz)
Brain activity	EEG (8 electrodes, 256 Hz)
Jaw muscle activity	EMG on lower jaw (3 electrodes, 256 Hz)
Eye activity	EOG (2 electrodes, 256 Hz)
Full body motion	Accelerometer at chest (32 Hz)

**Table 3 sensors-24-03548-t003:** Combined breathing detection results for all subjects. The overall ratio of time windows with detected breathing activity based on the PSG system was 93.9%. (PSG: Polysomnography; BPM: Breaths per minute).

	Radar at Bed Foot	Radar at Nightstand	Radar at Ceiling
Portion of time windows with breathing in radar data	98.5%	97.5%	81.4% *
Portion of time windows with breathing in radar **and** PSG data	93.0%	92.4%	77.3% **
Breathing frequency difference to PSG sensor within ±3 BPM	98.5%	98.3%	98.9%
Breathing frequency difference to PSG sensor within ±1 BPM	93.2%	92.8%	93.8%
Mean Absolute Percent Error of frequency difference	2.24%	2.36%	2.04%

* Excluding subject 11, for whom no breathing was detected from the ceiling radar: 97.8%. ** Excluding subject 11, for whom no breathing was detected from ceiling radar: 92.8%.

**Table 4 sensors-24-03548-t004:** Combined heartbeat detection results for all subjects. A heart rate based on the ECG data is derived in 92.6% of time windows. (BPM: Beats per minute).

	Radar at Bed Foot		Radar at Nightstand		Radar at Ceiling	
	All	Subset *	All	Subset *	All	Subset *
Portion of time windows with pulse in radar data	70.6%	81.4%	59.6%	66.1%	60.0%	82.3%
Portion of time windows with pulse in radar **and** PSG data	67.4%	77.6%	56.7%	62.7%	59.8%	82.1%
Heartbeat pulse frequency difference to PSG sensor within ±5 BPM	92.3%	95.5%	87.7%	88.6%	76.1%	82.0%
Heartbeat pulse frequency difference to PSG sensor within ±1 BPM	58.2%	60.4%	37.2%	37.6%	32.2%	35.2%
Mean Absolute Percent Error of frequency difference	3.25%	2.04%	4.09%	3.53%	6.09%	4.18%

* Excluding subject 09 from all positions and subject 11 from the ceiling position.

## Data Availability

Data sharing is not applicable due to concerns for the human subjects’ privacy. The source code of this article cannot be published.

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
