# Peer review of "Evaluation of Lateral Radar Positioning for Vital Sign Monitoring: An Empirical Study"

_sensors, 2024, doi:10.3390/s24113548_

Round 1
Reviewer 1 Report
Comments and Suggestions for Authors
In the paper Authors investigated and compared 3 different locations of the microwave radar locations for respiration and heart rate monitoring during sleep. The research results can be directly applied for sleep apnea and other slep disorders monitoring, as this is important part towards monitoring of person at home. The major benefit of such technique is complete non-invasiveness as no sensors have to be located on the patioent, which might cause problems with normal sleep.
Authors proposed alternative to classic Doppler approach - they use FMCW technique, which allows to differentiate and focus on certain regions of interest. Thus device can be positioned at certain angle at the end of the bed. Thus it can differentiate between regions giivng better sensitivity for respiration when focusing on the patient's chest.
Authors in clear way explained technique, they use, setup, experiment and compared with reference methods as well.
I hust want to congratulate authors and I thing paper can be published in present form.
Author Response
Dear Reviewer,
Thank you for your feedback to our manuscript! We hope that we have been able to improve the text.
Please see the attachments to find our detailed response to your remarks as a PDF with the name "Reviewer 1 - Author Response".
The color coding of the highlights is as follows.
- blue wavy underlined = text was added
- red crossed out = text was removed
- plain black text = text was kept the same way
Kind regards

Reviewer 2 Report
Comments and Suggestions for Authors
The paper describes the experimental research where the contactless measurement of human breathing and heartbeat signals is conducted by means of the radiofrequency radar exploiting FMCW signal. The strength of the paper consists in the real-life experiment with 33-hour long data recorded and further compared to the data gathered by a versatile polysomnography system which was considered as reference.
General remarks
1. The title of the paper does not correspond to its actual content. Thus, having read this title, a reader definitely expects that the paper will present the analysis of the problem of optimal, or at least more efficient, positioning of the radar sensor. In other words, the authors will prove analytically or by means of simulation that there is better placement for the sensors and show the technique how it has been achieved and, possibly, how someone else could achieve it in a similar problem.
Instead, what is presented in the paper is just a case study for one seemingly arbitrary chosen set of positions with rather a weak expert-like justification.
The title must be changed in order not to mislead readers!
2. The paper does not contain a proper description of the essential points such as the emitted signal model, radar signal processing methods, procedures for signal-of-interest extraction after the optimal receiver, etc. The attempt to describe how the data were processed are given in Section 3 with no a single formula!
3. Some incoherency in the structure seems to be in the paper. Hence, what the authors tried to described in Section 3 seems to be the description of the method, which is typically to be written in ‘Materials and methods’ Section, rather than ‘Data evaluation’.
4. The review of state-of-art provided in Introduction is recommended to be expanded since the authors seems to have missed some contemporary paper on the topic where the techniques for cardiac and respiration signals are investigated:
https://doi.org/10.1186/1687-1499-2011-20
https://doi.org/10.3390/inventions7030079
https://doi.org/10.1109/TBME.2021.3066876
https://doi.org/10.1109/ACCESS.2022.3190902
https://doi.org/10.1109/BioSMART54244.2021.9677856
Specific remarks
1. In line 63, the range resolution is estimated as large as 5 cm. It is not properly explained how it is possible to measure accurately the displacement of the order of 0.02 mm as it is shown with graphs in Fig. 3.
2. The reviewer could not understand what is the actual size of vital signs is exploited in the research. To be more precise, is it limited to the average frequencies of breathing and heartbeat process?
3. The design of the conducted experiment(s) is not thoroughly described. Thus, In subsection 2.2, it is written (line 180) that only data acquired from six participants are used. However, in Tables 3 and 4 and at the beginning of Section 4, some ‘Subjects’, 9 and 11, unexpectedly turn up. This raises the set of question, such as,
- how many people were actually involved, how different they were,
- what criteria for their selection was implied and why only healthy participants (except the inappropriate story with one) were engaged,
- how the data were aggregated: subject-wise or just a ‘bag of measurements’ regardless people they belong to,
and some others.
Critical remark
The paper does not reveal any results which are scientifically novel. Instead, the paper makes an impression of a case study where well-known techniques previously published are applied to a single experiment.
Summary
Taking into consideration the gravity of the remarks highlighted above, the reviewer cannot recommend this paper for publication in present form. However, the practical results might be interesting to many readers providing the paper has been properly rewritten. The resubmission may well be encouraged.
Comments on the Quality of English LanguageThe quality of English Language is rather high. An additional proof-reading is recommended for checking some punctuation marks and considering some simpler replacement for such words as ‘necessitating’ in line 106.
Author Response
Dear Reviewer,
Thank you for your feedback to our manuscript! We hope that we have been able to improve the text.
Please see the attachments to find our detailed response to your remarks as a PDF with the name "Reviewer 2 - Author Response".
The color coding of the highlights is as follows.
- blue wavy underlined = text was added
- red crossed out = text was removed
- plain black text = text was kept the same way
Kind regards

Reviewer 3 Report
Comments and Suggestions for Authors
This manuscript proposed a lateral positioning at the head or feet to detect breathing and heart rate activity during sleep. Although the proposed approach is a study with practical significance, the following issues should be considered and further explained.
1. The radar is deployed on the lateral position of the bed is not novel in this application. Actually, some articles and companies have already studies the position of radar. Such as the video in https://www.qingleitech.com/, the vital radar products is deployed at the head of the bed. For the reader, the significance of this manuscript is to explain the theory reasons of this deployment design. Thus the novelty of this manuscript is slight.
2. Although the manuscript proposed the methods of selection range bins and processing algorithm, it is hard to be convinced that there is no comparison with SOTA algorithms.
3. Furthermore, would more bins have been selected lead to more computational complexity? Whether the manuscript considers the tradeoff between performance and complexity.
4. In the experiment, the distance between radar 3 and the bed is farther than radar 1. Could this lead to a decrease in the accuracy of vital signs detection due to the more severe signal attenuation of radar 3?
5. In b) of the range bin selection algorithm, if the subject has random body motion, will this make the phase difference of all range bins exceed the threshold of 1mm? What to do if they all exceed the threshold?
6. In both respiration and heartbeat detection, just a different range bin is selected for each. However, it will be interesting to see what happens when multiple range bins are selected.
Comments on the Quality of English LanguageThe English description is generally understandable.
Author Response
Dear Reviewer,
Thank you for your feedback to our manuscript! We hope that we have been able to improve the text.
Please see the attachments to find our detailed response to your remarks as a PDF with the name "Reviewer 3 - Author Response".
The color coding of the highlights is as follows.
- blue wavy underlined = text was added
- red crossed out = text was removed
- plain black text = text was kept the same way
Kind regards

Round 2
Reviewer 2 Report
Comments and Suggestions for Authors
All the questions that were raised in my initial review have been properly addressed and necessary corrections have been made to the paper. The quality of the paper has significantly been improved.
Reviewer 3 Report
Comments and Suggestions for Authors
All my concerns have been addressed.
Comments on the Quality of English LanguageWriting is acceptable.